# Prophylactic Instillation of Hydrogen-Rich Water Decreases Corneal Inflammation and Promotes Wound Healing by Activating Antioxidant Activity in a Rat Alkali Burn Model

**DOI:** 10.3390/ijms23179774

**Published:** 2022-08-29

**Authors:** Momoko Kasamatsu, Takeshi Arima, Toyo Ikebukuro, Yuji Nakano, Yutaro Tobita, Masaaki Uchiyama, Akira Shimizu, Hiroshi Takahashi

**Affiliations:** 1Department of Ophthalmology, Nippon Medical School, Bunkyo-ku, Tokyo 113-8603, Japan; 2Department of Analytic Human Pathology, Nippon Medical School, Bunkyo-ku, Tokyo 113-8603, Japan

**Keywords:** hydrogen, prophylactic effect, corneal inflammation, antioxidative effect, SOD1, PGC-1α, antioxidant, alkali burn

## Abstract

Many studies have demonstrated the therapeutic effects of hydrogen in pathological conditions such as inflammation; however, little is known about its prophylactic effects. The purpose of this study is to investigate the prophylactic effects of hydrogen-rich water instillation in a rat corneal alkali burn model. Hydrogen-rich water (hydrogen group) or physiological saline (vehicle group) was instilled continuously to the normal rat cornea for 5 min. At 6 h after instillation, the cornea was exposed to alkali. The area of corneal epithelial defect (CED) was measured every 6 h until 24 h after alkali exposure. In addition, at 6 and 24 h after injury, histological and immunohistochemical observations were made and real-time reverse transcription polymerase chain reaction (RT-PCR) was performed to investigate superoxide dismutase enzyme (SOD)1, SOD2, and peroxisome proliferator-activated receptor gamma coactivator 1-alpha (PGC-1α) mRNA expression. CED at 12 h and the number of inflammatory infiltrating cells at 6 h after injury were significantly smaller in the hydrogen group than the vehicle group. Furthermore, SOD1 expression was significantly higher in the hydrogen group than the vehicle group at both 6 and 24 h, and the number of PGC-1α-positive cells was significantly larger in the hydrogen group than the vehicle group at 6 h after injury. In this model, prophylactic instillation of hydrogen-rich water suppressed alkali burn-induced inflammation, likely by upregulating expression of antioxidants such as SOD1 and PGC-1α. Hydrogen has not only therapeutic potential but also prophylactic effects that may suppress corneal scarring following injury and promote wound healing.

## 1. Introduction

Hydrogen reacts directly with free radicals in a redox reaction, particularly with hydroxyl radicals, which induces various therapeutic effects [1,2,3,4,5,6,7]. Unlike other common anti-inflammatory drugs, hydrogen has no side effects [2,8]. Recently, a new mechanism for hydrogen’s effects has been revealed. Hydrogen works not only through a direct pathway as described above but also through indirect pathways to upregulate antioxidants [9,10]. We reported that the expression of superoxide dismutase enzyme (SOD)1 was upregulated by continuous administration of hydrogen in rat corneal epithelial tissue [11]. Based on this fact, we hypothesized that hydrogen not only has therapeutic effects but may also work prophylactically through upregulation of antioxidants including SOD1. SOD is an enzyme that reacts with superoxide directly and also works as a nuclear transcription factor to reduce oxidative stress [12,13]. There are three subtypes of SOD: SOD1, SOD2, and SOD3. SOD1 is the main subtype and accounts for 80% of all SODs. In the present study, we prophylactically administered hydrogen-rich water onto the cornea of rats prior to corneal injury and investigated suppression of inflammation and upregulation of antioxidants.

## 2. Results

### 2.1. Antioxidant Activity of Normal Cornea after Hydrogen Instillation

The antioxidant activity of the normal rat cornea at 6 h after hydrogen instillation for 5 min was evaluated. SOD1, SOD2, and peroxisome proliferator-activated receptor gamma coactivator 1-alpha (PGC-1α) protein expression was evaluated by immunostaining and mRNA expression was measured by the real-time reverse transcription polymerase chain reaction (RT-PCR, Figure 1). SOD1 was detected mainly in the cytoplasm of corneal epithelial cells. On the other hand, PGC-1α was expressed in both nuclei and cytoplasm. Immunostaining revealed that the number of SOD1 (Figure 1e) and PGC-1α-positive cells (Figure 1g) in the hydrogen group was larger compared to the vehicle group. RT-PCR analysis supported this result; SOD1 (Figure 1h) and PGC-1α mRNA expression (Figure 1i) in the hydrogen group were significantly higher compared to those in the vehicle group. Furthermore, to evaluate trends in inflammation and anti-inflammation, mRNA expression of nuclear factor-kappa B (NF-κB), nuclear factor of kappa light polypeptide gene enhancer in B-cells inhibitor, alpha (Iκ-Bα), interleukin-10 (IL-10), and vascular endothelial growth factor (VEGF)-A were also examined using RT-PCR (Figure 1j–m). There were no significant differences in mRNA expression levels of NF-κB, Iκ-Bα, and IL-10; however, VEGF-A mRNA expression was significantly lower in the hydrogen group than in the vehicle group (Figure 1m).

### 2.2. Corneal Wound Healing after Alkali Burn

Images of fluorescein-stained corneas were captured every 6 h until 24 h after injury, and corneal epithelial defect (CED) areas were compared between the hydrogen and vehicle groups. (Figure 2). The CED began to decrease at 6 h after injury and disappeared by 24 h (Figure 2a–f). The CED area was significantly smaller in the hydrogen group than in the vehicle group at 12 h after injury (Figure 2g).

### 2.3. Inflammation and Oxidative Stress after Alkali Burn

The anti-inflammatory effect of prophylactic instillation of hydrogen was evaluated (Figure 3 and Figure 4). We performed naphthol AS-D chloroacetate esterase (EST) staining for neutrophil infiltration (Figure 3a,b), ED-1 immunostaining for pan macrophages (Figure 4a,b), and ED-2 immunostaining for M2 macrophages (Figure 4c,d). Furthermore, RT-PCR analysis of NF-κB, Iκ-Bα, IL-10, and VEGF-A was also performed (Figure 5). EST-positive cells were found in the corneal limbus at 6 h after injury, which is thought to be the early phase of inflammation, then neutrophil infiltration to the corneal center was observed at 24 h after injury (Figure 3a,b). The number of neutrophils was significantly lower in the hydrogen group than in the vehicle group at 6 h in the limbus (Figure 3c). The number of ED-1-positive cells was significantly smaller in the hydrogen group in the central cornea and limbus at 24 h after injury compared to the vehicle group (Figure 4e), while no significant differences were observed in ED-2-positive cells at the same time points (Figure 4f). These results suggested that M1 macrophages were suppressed in the hydrogen group. RT-PCR analysis revealed that mRNA expression of NF-κB (Figure 5a) and Iκ-Bα (Figure 5b) was significantly lower in the hydrogen group than in the vehicle group at 6 h after injury. Whereas, mRNA expression of IL-10 was higher (Figure 5c) and VEGF-A was lower (Figure 5d) at 24 h after injury in the hydrogen group. The anti-oxidative stress effect of prophylactic instillation of hydrogen was also evaluated. We performed anti-8-Hydroxy-2′-deoxyguanosine (8-OHdG) immunostaining for detecting oxidative stress (Figure 3d–g). 8-OHdG staining is positive both in the infiltrating stromal cells and the epithelial cells in the corneal limbus after alkali injury. The 8-OHdG positive cells were apparent both in the hydrogen group and the vehicle group.

### 2.4. Antioxidant Activity after Alkali Burn

To investigate the antioxidative activity of prophylactic hydrogen instillation, we performed immunostaining using SOD1, SOD2, and PGC-1α antibodies (Figure 6a–h). At 6 h after injury, the numbers of SOD1 (Figure 6i) and PGC-1α-positive cells (Figure 6k) in the corneal limbus were significantly higher in the hydrogen group compared to the vehicle group. No positive cells were found in the corneal center in all eyes due to CED. At 24 h after injury, the number of SOD1-positive cells in both the corneal limbus and the corneal center was higher in the hydrogen group than in the vehicle group (Figure 6i). These results suggest that pre-injury hydrogen instillation increased SOD1 and PGC-1α in the corneal epithelium. Furthermore, we performed RT-PCR analysis of SOD1, SOD2, and PGC-1α (Figure 6l–n). Higher mRNA expression levels of SOD1 were found in the hydrogen group than in the vehicle group at 6 and 24 h after injury (Figure 6l). These results coincided and supported the results of immunostaining.

## 3. Discussion

Corneal alkali injury is one of the most serious conditions in the field of ophthalmology, causing corneal epithelial defects, inflammation, neovascularization, and corneal opacity, which can result in permanent vision loss. In this study, we employed a rat alkali burn model widely used for investigating corneal wound healing and inflammatory responses to corneal damage [14,15]. We previously reported that hydrogen-rich water instillation has antioxidant and anti-inflammatory effects on corneal alkali injury [11] and demonstrated the therapeutic effects of hydrogen. However, its prophylactic effect has not yet been clearly elucidated. In this study, we demonstrated that hydrogen instillation prior to corneal alkali burn shows anti-inflammatory and antioxidant effects and promotes corneal wound healing.

Immunostaining showed a significantly smaller number of infiltrating neutrophils in the hydrogen group compared to the vehicle group in the corneal limbus at 6 h after corneal injury. To assess the inflammation from a different perspective, we measured the activities of NF-κB and Iκ-Bα. NF-κB plays a major role in regulating the expression of many genes related to cell growth, inflammation, and apoptosis. Iκ-Bα is a suppressor protein that binds to NF-κB. When Iκ-Bα binds to NF-κB, it inhibits NF-κB transfer into the nucleus and restricts the NF-κB signaling pathway [16]. In the present study, administration of hydrogen downregulated the mRNA expression of NF-κB and Iκ-Bα in the cornea at 6 h after alkali exposure. These findings suggest that the pre-administered hydrogen instillation has an anti-inflammatory effect against subsequent alkali injury at the early stage of inflammation. Downregulated Iκ-Bα mRNA expression may be induced through negative feedback from reduced NF-κB activity, which was suppressed by hydrogen. Additionally, corneal edema after alkali damage seems more likely to occur in the vehicle group compared to the hydrogen group by hydrogen’s suppressing inflammation and fibroblasts, which are shown in Figure 1a–d and Figure 4a–d. Furthermore, an additional experiment of 8-OHdG staining was performed to evaluate the anti-oxidative stress effect of prophylactic instillation of hydrogen. After alkali injury, 8-OHdG-positive cells were observed both in the infiltrating stroma and in the epithelium in the cornea, and the number of them is apparently greater in the hydrogen group compared in the vehicle group, suggesting the involvement of free radicals in the mechanism of prophylactic hydrogen’s instillation.

We observed the effect of prophylactic hydrogen instillation on macrophage infiltration. Macrophages are classified into M1 and M2 macrophages, and M1 macrophages are involved in inflammatory development. On the contrary, M2 macrophages have a role in immunosuppression and tissue repair at the late stage of inflammation. The balance of macrophage phenotypes is also important in determining the degree of inflammation in addition to macrophage infiltration [17]. Topical hydrogen treatment for the alkali-injured cornea has been reported to increase M2 macrophages and contribute to anti-inflammatory effects in corneal wound healing [11]. In the present study, a lower number of ED-1-positive cells (M1 macrophages) was found in the hydrogen group than in the vehicle group, while no significant difference was observed in ED-2-positive cells (M2 macrophages) between the two groups. In previous reports, M1 macrophages were dominant at days 1 to 3 [18,19,20] after the start of wound healing, and M2 macrophages began to increase from days 3 to 5 [18,19,20]. We focused on the early stage of inflammation in order to evaluate the prophylactic effect of hydrogen in this study, which may be the reason why no significant difference in the number of ED-2-positive cells was observed.

Hydrogen not only reacts directly with free radicals to exhibit antioxidant activity, but also shows its effect through indirect pathways. Various factors such as SOD1 and PGC-1α are thought to be involved in these indirect pathway [5,21]. PGC-1α is a transcription coactivator that regulates the expression of target genes by binding to various transcription factors including nuclear receptors, and it has a role in cell energy production [22,23]. PGC-1α has been known to reduce reactive oxygen production [24]. In this study, immunostaining and RT-PCR showed that hydrogen instillation onto the normal rat cornea increased SOD1 and PGC-1α activities in the corneal epithelial tissue. This fact means that SOD1 and PGC-1α are activated by hydrogen and remained on the cornea for more than 6 h after hydrogen administration ceased. Furthermore, in this study, loading of prophylactic hydrogen before alkali injury increased the number of SOD1- and PGC-1α-positive cells in the corneal epithelial limbus at 6 h after injury, and increased the number of SOD1-positive cells at 24 h after injury. In addition, RT-PCR analysis showed that prophylactic instillation of hydrogen increased SOD1 mRNA expression in the cornea at 6 and 24 h after injury, which supported the immunostaining results. The difference in PGC-1α expression between immunostaining and RT-PCR may be due to sample differences; the corneal epithelium was applied for immunohistochemistry, whereas the entire cornea was employed for PCR.

Finally, a low level of VEGF mRNA expression was observed with prophylactic hydrogen-rich water instillation. VEGF plays an essential role in inflammation [14]. In age-related macular degeneration (AMD), oxidative stress is thought to play an important role [25]. The fact that a mouse model that lacks the SOD1 gene shows AMD-like characteristics in the retina [26,27] supports oxidative stress as a major factor in AMD. Anti-VEGF therapy is clinically accepted worldwide as the gold standard for AMD today. Interestingly, in a mice model, hydrogen administration in AMD disease was found to reduce oxidative stress and lower VEGF expression [7]. Further investigation is needed with regard to the prophylactic use of hydrogen instillation in ophthalmology.

## 4. Materials and Methods

### 4.1. Animals and Ethics Statement

Eight-week-old Wistar rats from Sankyo Laboratory Service (Tokyo, Japan) were used for all experiments in this study. All rats were maintained in a filtered-air laminar-flow enclosure in a specific pathogen-free facility and were provided food *ad libitum*. All animal experiments were conducted in compliance with the Experimental Animal Ethics Review Committee of Nippon Medical School (approval number: 2021-034, 1 April 2021) and all procedures conformed to the Association for Research in Vision and Ophthalmic and Visual Research.

### 4.2. Administration of Hydrogen to Normal Cornea

Under general isoflurane anesthesia, hydrogen-rich water (hydrogen group) or physiological saline (vehicle group) was instilled continuously onto the right cornea of rats for 5 min. Hydrogen-rich water (Melodian Co., Ltd., Osaka, Japan) containing hydrogen concentrations from 1.2 to 1.6 ppm was used at room temperature in all experiments. At 6 h after instillation, rats were sacrificed by exsanguination under isoflurane anesthesia. Pathological and molecular biological analyses were performed on the enucleated eyes.

### 4.3. Prophylactic Administration of Hydrogen and Alkali Burn Model

Hydrogen-rich water or physiological saline was instilled continuously onto the right cornea of rats for 5 min as stated previously. At 6 h after instillation, alkali burn was generated at the cornea under general anesthesia (inhalation of isoflurane). A round filter paper 3.2 mm in diameter and impregnated with 1 N NaOH was applied to the center of the cornea for 1 min, followed by irrigation with 40 mL of physiological saline. At 6 and 24 h after injury, rats were sacrificed by exsanguination under isoflurane anesthesia. Pathological and molecular biological analyses were performed on the enucleated eyes.

### 4.4. Measurement Area of Corneal Epithelial Defect

Images of the fluorescein-stained cornea were captured every 6 h until 24 h after injury using an ophthalmic operating microscope (Leica M820 F19, Leica Microsystems, Wetzlar, Germany) to evaluate the corneal wound healing process. ImageJ software (version 1.52, National Institutes of Health; http://rsb.info.nih.gov/ij/ (accessed on 15 December 2020) was used to analyze the images and calculate the total corneal area and the area of CED.

### 4.5. Histological and Immunohistochemical Analyses

The excised eyes were fixed in 10% buffered formalin and embedded in paraffin. Corneal sections (3 μm thick) were cut from the paraffin blocks and placed onto precoated slides. HE staining and EST staining with HE was performed for deparaffinized tissues to detect infiltrating neutrophils. Primary antibodies against the following antigens were used for immunohistochemistry: anti-rat CD68 (ED-1, 1:500; BMA, Nagoya, Japan) and CD163 (ED-2, 1:50; BMA, Nagoya, Japan) antibodies for the detection of macrophages (ED-1 for M1 macrophages and ED-2 for M2 macrophages), anti-8-OHdG antibody (JaICA, Shizuoka, Japan) for the detection of DNA oxidative stress, SOD1 (1:200; ab13498; Abcam, Cambridge, MA, USA), SOD2 (1:50; ab13533, Abcam, Cambridge, MA, USA) and PGC-1α (1:50; ab54481; Abcam, Cambridge, MA, USA) for antioxidant markers. Three regions in the central cornea and two regions in the limbus were observed at a magnification of 400×. The number of positively stained cells was counted and the average number was calculated.

### 4.6. Real-Time RT-PCR

Corneas were carefully isolated from enucleated eyes. Tissue samples were stabilized in RNAlater (Qiagen, GmbH, Hilden, Germany). Total RNA was extracted using an extraction reagent (ISOGEN II; Nippon Gene, Tokyo, Japan). RNA concentrations were measured using a NanoDrop ND1000 V3.2.1 spectrophotometer (Thermo Fisher Scientific, Waltham, MA, USA). cDNA was prepared using a High-Capacity cDNA Reverse Transcription Kit with RNase Inhibitor (Applied Biosystems, Foster City, CA, USA). The RT-PCR assay was performed using the QuantStudioTM3 Real-Time PCR System (Thermo Fisher Scientific, Waltham, MA, USA), THUNDERBIRD SYBR qPCR Mix (TOYOBO, Osaka, Japan), and specific primers. The primer sequences used in this study are shown in Table 1. The expression level of each target gene was normalized to β-actin.

### 4.7. Statistical Analysis

Statistical analysis was performed using an unpaired Student’s *t* test. All results are expressed as the mean ± standard error, and *p* < 0.05 was considered to indicate statistical significance.

## 5. Conclusions

Prophylactic instillation of hydrogen suppressed alkali burn-induced oxidative stress of the cornea and promoted wound healing. Hydrogen not only has therapeutic potential but also preventive effects. The administration of hydrogen in advance may be useful in protecting the eyes from damage which occurs during scheduled procedures such as intraocular surgery.

## Figures and Tables

**Figure 1 ijms-23-09774-f001:**
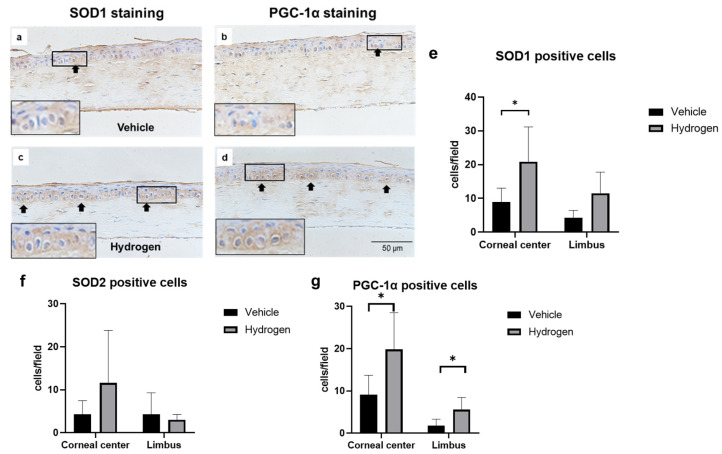
Effects of hydrogen in the normal rat cornea. Expression of SOD1 ((**a**): vehicle group, (**c**): hydrogen group) and PGC-1α ((**b**): vehicle group, (**d**): hydrogen group) in the normal cornea at 6 h after application of hydrogen-rich water. SOD1 was strongly expressed in the cytoplasm in the hydrogen group (**a**), whereas low levels were expressed in the cytoplasm in the vehicle group (**c**). PGC-1α was strongly expressed in the nuclei and cytoplasm in the hydrogen group (**b**), whereas low levels were expressed in the nuclei and cytoplasm in the vehicle group (**d**). Numbers of SOD1-positive cells (**e**), SOD2-positive cells (**f**) and PGC-1α-positive cells (**g**) in the normal cornea at 6 h after hydrogen instillation in the central and peripheral cornea. The number of SOD1-positive cells in the central cornea was significantly higher in the hydrogen group than the vehicle group (**e**). The number of PGC-1α-positive cells was also higher both in the central and peripheral cornea (**g**). Data are presented as the mean ± standard error (*n* = 6 per group). mRNA expression of molecules related to antioxidants and inflammation including SOD1 (**h**), PGC-1α (**i**), NF-κB (**j**), Iκ-Bα (**k**), IL-10 (**l**), and VEGF-A (**m**). mRNA expression of SOD1 (**h**) and PGC-1α (**i**) was significantly higher in the hydrogen group compared to the vehicle. No significant differences were found in the levels of NF-κB (**j**), Iκ-Bα (**k**) and IL-10 (**l**), whereas VEGF-A mRNA expression was notably higher in the hydrogen group (**m**). Hydrogen-rich water instillation resulted in antioxidant activation in healthy rat corneal tissue. Data are presented as the mean ± standard error (*n* = 9 per group). * *p* < 0.05.

**Figure 2 ijms-23-09774-f002:**
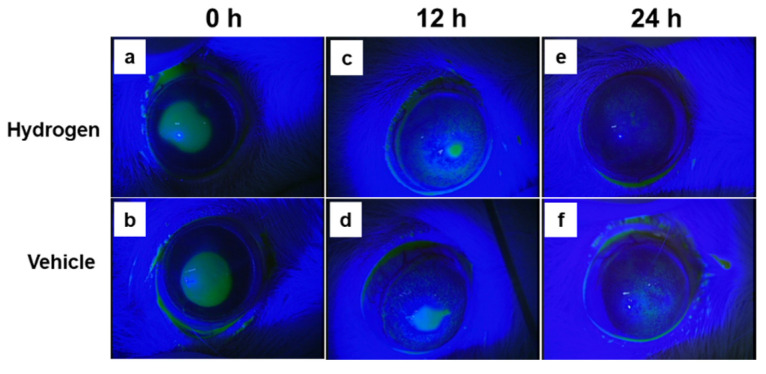
Corneal wound healing after alkali burn. Images of the fluorescein-stained cornea ((**a**,**b**): immediately after injury, (**c**,**d**): 12 h after injury, (**e**,**f**): 24 h after injury). The ratio of CED area to total corneal area (**g**). The time-dependent change in the vehicle group ((**h**,**i**): central region, (**l**,**m**): peripheral region) and the hydrogen group ((**j**,**k**): central region, (**n**,**o**): peripheral region). CED ratio at 12 h after injury was smaller in the hydrogen group than in the vehicle group (**g**). * *p* < 0.05. Data are presented as the mean ± standard error (*n* = 5 per group).

**Figure 3 ijms-23-09774-f003:**
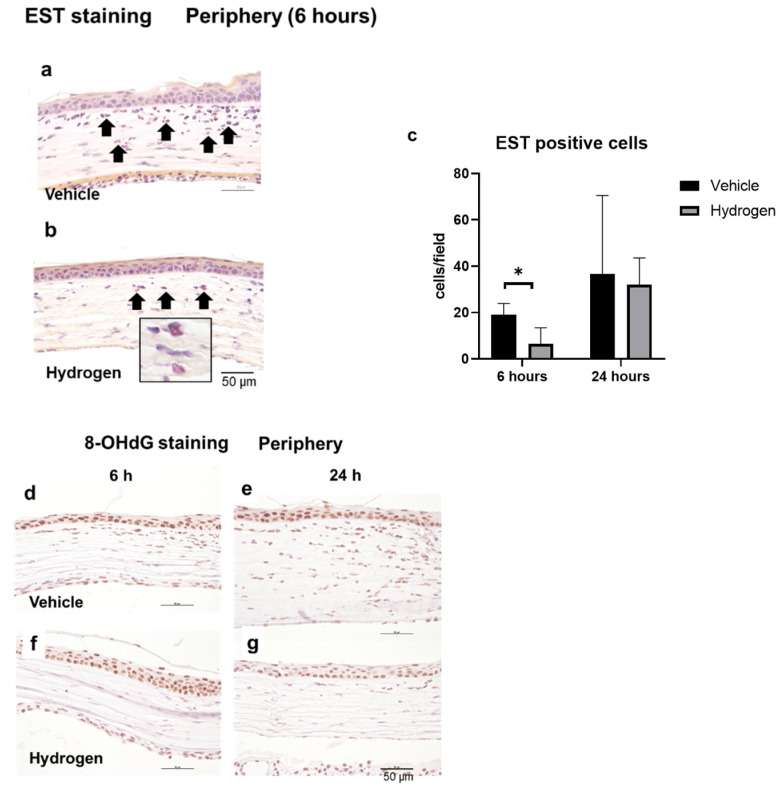
Effects of hydrogen on neutrophil infiltration and oxidative stress. EST staining was performed to evaluate neutrophil infiltration in the vehicle group (**a**) and the hydrogen group (**b**) in the corneal limbus at 6 h after injury (black arrows: EST-positive cells); 8-OHdG staining was also performed in the vehicle group ((**d**): 6 h, (**e**): 24 h) and in the hydrogen group ((**f**): 6 h, (**g**): 24 h) in the corneal limbus. Bar, 50 μm. The number of EST-positive neutrophils at 6 h after injury was lower in the hydrogen group than in the vehicle group (**c**). * *p* < 0.05. Data are presented as the mean ± standard error (*n* = 5 per group).

**Figure 4 ijms-23-09774-f004:**
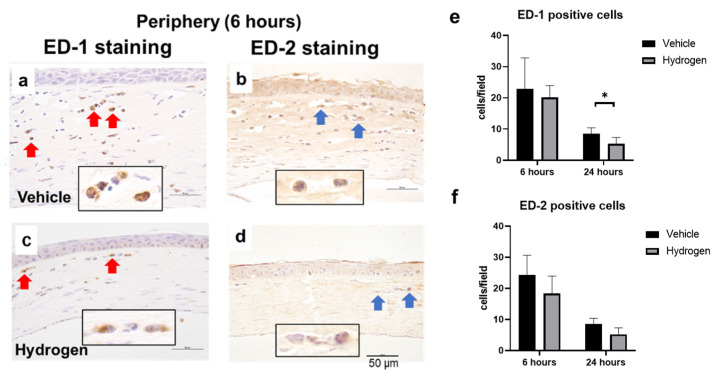
Effects of hydrogen on macrophage infiltration. ED-1 (red arrow) and ED-2 (blue arrow) positive cells in the peripheral cornea at 6 h after injury (**a**,**b**): vehicle group, (**c**,**d**): hydrogen group). The number of ED-1-positive (**e**) and ED-2-positive (**f**) macrophages. Hydrogen application significantly suppressed ED-1-positive macrophages at 24 h after injury compared to the vehicle group. No significant differences were found in the number of ED-2-positive macrophages. Data are presented as the mean ± standard error (*n* = 5 per group). * *p* < 0.05. Scale bar: 50 μm.

**Figure 5 ijms-23-09774-f005:**
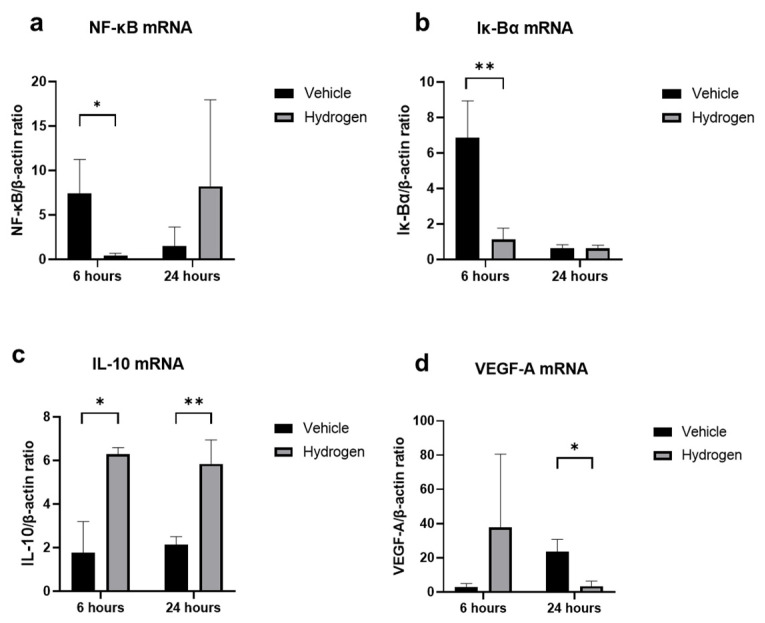
mRNA expression of inflammation-related molecules. Real-time RT–PCR was performed to evaluate mRNA expression of NF-κB (**a**), Iκ-Bα (**b**), IL-10 (**c**), and VEGF-A (**d**) in the cornea at 6 and 24 h after injury. The mRNA expression levels of NF-κB, Iκ-Bα, and IL-10 were significantly higher in the hydrogen group compared to the vehicle group at 6 h (**a**,**b**), and VEGF-A levels were also remarkably higher in the hydrogen group at 24 h after injury. Instillation of hydrogen-rich water significantly suppressed NF-κB, Iκ-Bα, and VEGF-A mRNA expression and upregulated IL-10 at the early stage of inflammation. Data are presented as the mean ± standard error (*n* = 5 per group). * *p* < 0.05, ** *p* < 0.01.

**Figure 6 ijms-23-09774-f006:**
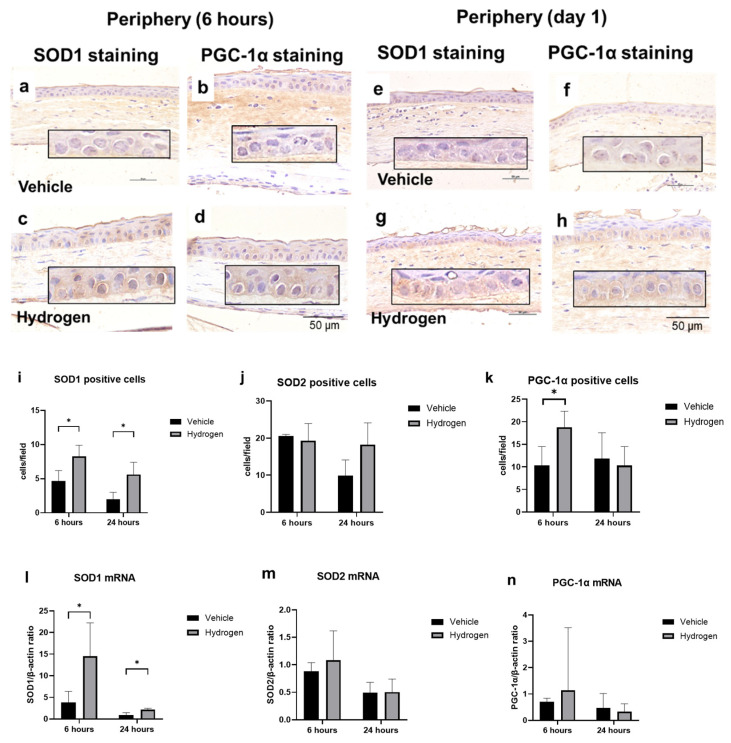
Effects of prophylactic hydrogen administration on antioxidant activity. Immunohistochemistry of SOD1 ((**a**,**e**): vehicle group at 6, 24 h, (**c**,**g**): hydrogen group at 6, 24 h, respectively) and PGC-1α ((**b**,**f**): vehicle group at 6, 24 h, (**d**,**h**): hydrogen group at 6, 24 h, respectively) in the corneal tissue (instilled prophylactically with hydrogen) after injury. The numbers of SOD1- (**i**), SOD2- (**j**), and PGC-1α- (**k**) positive cells at 6 and 24 h in the limbal cornea after injury. The number of SOD1-positive cells was significantly higher in the hydrogen group at both 6 and 24 h after injury in the peripheral cornea compared to the vehicle group (**i**). The number of PGC-1α-positive cells was also higher in the hydrogen group at 6 h after injury in the peripheral cornea compared to the vehicle group (**k**). Real-time RT–PCR was performed to measure SOD1 (**l**), SOD2 (**m**), and PGC-1α (**n**) mRNA in the cornea at 6 and 24 h after injury. The mRNA expression of SOD1 was significantly higher in the hydrogen group than in the vehicle group (**l**). Prophylactic instillation of hydrogen significantly upregulated the activity of SOD1. Moreover, it may also have upregulated the activity of PGC-1α. Data are presented as the mean ± standard error (*n* = 5 per group). * *p* < 0.05.

**Table 1 ijms-23-09774-t001:** PCR primers used in this study.

Gene	Forward Primer Sequence (5′-3′)	Reverse Primer Sequence (5′-3′)
β-actin	GCAGGAGTACGATGAGTCCG	ACGCAGCTCAGTAACAGTCC
SOD1	AATGTGTCCATTGAAGATCGTGTGA	GCTTCCAGCATTTCCAGTCTTTGTA
SOD2	AGGGCCTGTCCCATGATGTC	AGAAACCCGTTTGCCTCTACTGAA
PGC-1α	GTGCAGCCAAGACTCTGTATGG	GTCCAGGTCATTCACATCAAGTT
NF-κB	TGGACGATCTGTTTCCCCTC	TCGCACTTGTAACGGAAACG
Iκ-Bα	TGACCATGGAAGTGATTGGTCAG	GATCACAGCCAAGTGGAGTGGA
IL-10	GCAGGACTTTAAGGGTTACTTGG	GGGGAGAAATCGATGACAGC
VEGF-A	GCAGCGACAAGGCAGACTAT	GCAACCTCTCCAAACCGTTG

## Data Availability

The data presented in this study are available on reasonable request from the corresponding author. The data are not publicly available due to privacy restrictions.

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
