# Peer review of "Prophylactic Instillation of Hydrogen-Rich Water Decreases Corneal Inflammation and Promotes Wound Healing by Activating Antioxidant Activity in a Rat Alkali Burn Model"

_ijms, 2022, doi:10.3390/ijms23179774_

Round 1
Reviewer 1 Report
In this manuscript Kasamatsu et. al. have nicely demonstrated the benefits of hydrogen therapy on corneal wound healing via its antioxidant properties. In addition, they found in this model that prophylactic instillation of hydrogen-rich water suppressed alkali burn-induced inflammation through upregulating expression of antioxidants such as SOD1 and PGC-1α. However, authors needs additional experiments to solidify their claims, I have few suggestions to improve the quality of the manuscript.
1. It is not clear in the manuscript that hydrogen-rich water enhances the activity of SOD1 and SOD2 as authors have not performed any enzymatic activity or measured ROS levels in healing corneal epithelium. Expression of antioxidant enzymes as such does not say much about their activities. It is recommended to investigate antioxidant activity and measurement of ROS in corneal epithelial cells pre-treated with or without hydrogen. ROS can be detected with DHE, or MitoSox dyes using FACS or spectrophotometer method.
2. In figure2, it is also recommended to show histology of healing cornea with hematoxylin and eosin staining at different time points.
3. Immunostainings in figure 2 and 3 is not clear, needs to show magnified view or perform immunofluorescence stainings with specific immune cell markers.
4. It not clear to me how cornea shrinks following hydrogen treatment as shown in figure 1a-d and figure 4a-d. please explain this phenomenon in the discussion.
5. It is also recommended to improve the quality of immunostainings in figure 6a-h as such they are not convincing, tissue looks overexposed in hydrogen treated group.
Reviewer 2 Report
Interesting and relevant article in the area of prophylactic Instillation of Hydrogen-Rich Water Decreases Corneal Inflammation and Promotes Wound Healing by Acti-vating Antioxidant Activity in a Rat Alkali Burn Model.
The authors investigate the prophylactic effects of hydrogen-rich water instillation in a rat corneal alkali burn model.
The selected topic of the article is original and relevant in the study. A study on the prophylactic and therapeutic instillation of hydrogen-rich water to suppress inflammation caused by alkali burn by enhancing the expression of antioxidants, the effects of corneal scarring following injury and promoting wound healing.
Such research is rare. The presented model is described in the article for the first time.
The presented research methods are modern and meet the purpose. Additional research is not required.
In the article, the conclusions with evidence and arguments logically follow from the presented study and correspond to the purpose.
References correspond to the topic and are logically justified
The figures are presented informatively and correspond to the description in the article.
Round 2
Reviewer 1 Report
Manuscript is now significantly improved and I would
recommend to accept this manuscript.